# The role of selective attention in the positivity offset: Evidence from event related potentials

Regard M. Booy[1]*, Patrick L. Carolan[2]

**1** Department of Psychology, Simon Fraser University, Burnaby, British Columbia, Canada, **2** Department of Psychology, Saint Mary's University, Halifax, Nova Scotia, Canada

\* rmb8@sfu.ca

**Data Availability Statement:** Data cannot be shared publicly because of the requirements of the university's Department of Ethics at the time. In accordance with the ethics approval of this study, all raw data containing participant responses was

## Abstract

Some research suggests that positive and negative valence stimuli may be processed differently. For example, negative material may capture and hold attention more readily than equally arousing positive material. This is called the negativity bias, and it has been observed as both behavioural and electroencephalographic (EEG) effects. Consequently, it has been attributed to both automatic and elaborative processes. However, at the lowest levels of arousal, faster reaction times and stronger EEG responses to positive material have been observed. This is called the positivity offset, and the underlying cognitive mechanism is less understood. To study the role of selective attention in the positivity offset, participants completed a *negative affective priming* (NAP) task modified to dissociate priming for positive and negative words. The task required participants to indicate the valence of a target word, while simultaneously ignoring a distractor. In experiment 1, a behavioural facilitation effect (faster response time) was observed for positive words, in stark contrast to the original NAP task. These results were congruent with a previously reported general categorization advantage for positive material. In experiment 2, participants performed the task while EEG was recorded. In additional to replicating the behavioural results from experiment 1, positive words elicited a larger Late Positive Potential (LPP) component on ignored repetition relative to control trials. Surprisingly, negative words elicited a larger LPP than positive words on control trials. These results suggest that the positivity offset may reflect a greater sensitivity to priming effects due to a more flexible attentional set.

## Introduction

Emotions likely evolved as an efficient way to determine the survival value of a stimulus [1]. The preferential processing of emotional material [2,3], facilitates allocation of cognitive resources towards evolutionarily important information. However, positive and negative valence material are treated differently in different situations [4]. For example, even when controlling for arousal, negative stimuli have a stronger influence on cognitive operations compared to positive stimuli [5]; [6,7]. Curiously, positive and negative valence stimuli (both words and images) are rated differently at high versus low levels of arousal. Specifically, positivity rating of positive valence stimuli are higher than negativity ratings of negative stimuli at

kept for a period of 5 years and then destroyed. Aggregate data was retained for data analysis purposes but cannot be shared outside of the research team who conducted the study. For questions regarding data accessibility and ethics approval of this study, contact the Department of Research Ethics at Simon Fraser University (dore@sfu.ca). Researchers who want to replicate this study can contact the Laboratory for Attention Memory and Perception (attnlab@sfu.ca) for access to the E-prime run files, instructions to participants, and instructions to research assistants.

**Funding:** The author(s) received no specific funding for this work.

**Competing interests:** The authors have declared that no competing interests exist.

low levels of arousal, but negativity ratings of negative stimuli are higher than positivity rating of positive stimuli at high levels of arousal [8,9]. In linear regression terms, when positive and negative valence are regressed on arousal, negativity has a steeper slope. This effect is referred to as the *negativity bias* and is thought to reflect an evolutionary need to selectively process potentially threatening aspects of the environment [10]. On the other hand, positivity has a higher intercept value. This effect is referred to as the *positivity offset* and seems to reflect a tendency toward positive evaluations in low emotionally arousing situations [11,12]. The positivity offset may have evolved to encourage approach behaviours that facilitate engagement with the environment when external stimulation is low [13].

The positivity offset is reliably observed in non-clinical populations under-low arousing experimental conditions. For example, healthy controls show a significantly higher positivity offset compared to currently depressed groups [12,14,15]. It often manifests as more positive evaluations of neutral images [8,12] but can also present as faster responses to positive material. Other studies examining event-related potentials (ERPs) show positive material is accompanied by larger amplitudes in some later components [14,15]. However, this is only seen when using low-arousing stimuli such as words [14,16,17]. While a fair bit is known about selective attention in the negativity bias, far less is known about the mechanisms underlying the positivity offset. It is possible that the positivity offset is more sensitive to experimental conditions since negative information has a stronger effect on cognitive processes [10,11]. The purpose of this study was to isolate the positivity offset in order to examine the underlying attentional mechanisms.

The negativity bias has been attributed to both automatic and controlled cognitive processes. One proposition is an early orienting bias reflecting an evolutionary need to quickly process negative information [18]. The result is greater attentional capture by negative material to facilitate a quick response to a potential threat. Another potential mechanism is delayed attentional disengagement. On many cognitive tasks (e.g. the emotional Stroop or eStroop task) the negativity bias manifests as slower responses to negative, than positive or neutral stimuli [3,19]. This slow-down has been explained as delayed disengagement from negative material that disrupts on going, task-relevant processes including response generation [3], which may allow for enhanced elaborative processing of negative material.

Electrophysiological investigations into the processing of emotional stimuli suggest involvement of both automatic and controlled cognitive processes. Early modulations (180-300ms) thought to reflect automatic, bottom-up aspects of attentional capture have been described over both posterior (early posterior negativity or EPN) [20–23], and anterior scalp (early anterior positivity or EAP) [22,24–26]. The EPN is observed as a negative deflection over temporo-occipital regions that peaks between 250-300ms post stimulus [17]. The EPN is sensitive to stimulus arousal [27], and its amplitude also varies with other factors such as picture content suggesting it reflect early selective attention to emotional stimuli [22]. The EAP, which presents as a positive deflection over frontal regions peaking between 200-300ms post stimulus [26], is comparatively less studied. More research is needed to differentiate the cognitive processes underlying the EAP from those related to the EPN. However, it is also sensitive to stimulus arousal [24]. Thus, early ERP modulations may represent attentional capture by emotional stimuli. Consistently, some studies report a negativity bias in these early components [2,23,24].

A negativity bias has also been reported in the late positive potential (LPP; also called the late positive complex or LPC [11]). For example, recent research has shown that when participants maintained a frown, LPP amplitude to negative pictures was increased [28]. The LPP is a broadly distributed, sustained wave over posterior scalp, peaking between 350-750ms [2]. LPP amplitude is affected by a range of factors including arousal, stimulus meaning, and task

demands [22]. It is thought to reflect the amount of working memory (WM) resources allocated to the maintenance of motivationally relevant material [22,26]. Thus, larger LPP amplitude indicates greater top-down, elaborative processing of a stimulus [29–31].

The electrophysiological data can be explained by a two-stage model of stimulus perception [22]. Limited cognitive resources must be allocated to the most relevant stimuli. According to these theories, early components such as the EPN or EAP may index the potential relevance of a stimulus. This signals to the system that more attentional resources should be devoted to processing that stimulus. In the second stage, the capacity-limited system allocates resources to the processing of stimuli. The amplitude of the LPP provides an index the amount of resources devoted to the stimulus. Thus, together these components show the allocation of selective attentional resources to emotional material.

Studies using emotional words often show a different pattern of results from studies using emotional pictures possibly because emotional words are inherently less arousing than pictures [16,32]. The mental representations of words are stored in semantic networks linked to all associated concepts, actions, and emotions [16,33]. Essentially, a written word is a symbol with semantic and affective connotations, rather than inherent meaning [34] making words less arousing than pictures. In these studies, the EPN is typically larger to both positive and negative compared to neutral words [17], and appears to be affected by arousal rather than valence [35,36]. The LPP shows more variable results. While LPP amplitude is generally larger for emotional compared to neutral words [37–40] it is also more susceptible to other factors such as task demands [36,37]. As a result, some studies report larger LPP amplitude to positive compared to negative and neutral words [14,16,17,32]. This is consistent with the positivity offset and may reflect the fact that words are inherently less arousing stimuli compared to pictures [16].

In these low arousing experimental conditions (i.e. a non-clinical sample, responding to low-arousing stimuli), selective attentional mechanisms involved in the positivity offset can be examined using the negative affective priming (NAP) task. *Negative priming* occurs when a response to a target is slowed because similar information was presented as a distractor during a prior stimulus display. This effect has been observed for target words presented after a semantically related distractor (i.e. negative semantic-priming [41]), but also for affective targets presented after an affectively congruent distractor (i.e. negative affective-priming) [42,43]. In the NAP task, participants are presented with a prime and a probe display in succession. Prime and probe displays each contain two stimuli, a *distractor* that participants are asked to ignore, and a *target* that they respond to by identifying its emotional valence as quickly as possible. Targets and distractors are distinguished based on simple physical characteristics, such as text colour (respond to blue words and ignore red words) or capitalization (respond to uppercase word and ignore lowercase words). Participant reaction time (RT) is analysed for probe displays only, based on the valence (positive or negative) of the prime-distractor (i.e. the distractor on the prime display), and the prime-distractor's congruence/incongruence with the probe-target (i.e. the target on the probe display) [42]. On Ignored Repetition (IR) trials, the valence of the prime-distractor is congruent with the valence of the probe-target. On control trials, the prime-distractor and probe-target valence are incongruent. To quantify the effect of previously ignoring a word of the same valence as the probe-target, the NAP effect is calculated as the RT difference between IR and the corresponding control trials for positive and negative valence. If the prime-distractor is effectively ignored, then responses to a subsequent probe-target of the same valence is delayed, resulting in a larger NAP effect. Thus, the size of the NAP effect is generally thought to measure the individual's ability to inhibit irrelevant emotional material of a particular valence from entering working memory [43–50]. By examining the NAP effect for positive material under conditions favouring the positivity offset, the

contribution of inhibitory processes to the positivity offset can be determined. The addition of event related potentials (ERPs) should reveal how the prime-distractor influences processing of the probe-target.

If a reduced ability to ignore task-irrelevant positive material in a low-arousal state underlies the positivity offset, then a reduced NAP effect to positive words should be observed. Whereas this reduced NAP should enhance the maintenance of the positive probe-target in WM, the reduced NAP effect should be accompanied by a larger LPP on IR trials. Two experiments were conducted to test these hypotheses. In experiment 1, a modified version of the NAP task designed to fully dissociate priming effects for positive and negative words was directly compared to the original version of the NAP task [43]. In experiment 2, participants completed the modified version of the task while electroencephalography (EEG) activity was recorded.

## Experiment 1

The original version of the NAP task [43] compared the NAP effect for positive and negative material. However, the design was unable to dissociate priming effects for positive and negative material. On IR trials for negative words, the prime slide contained a negative distractor and a positive target, while the probe slide contained the inverse; a negative target and a positive distractor. On control trials for negative words, both the distractor and target were positive, while the probe slide contained a negative target and a positive distractor. Thus, positive and negative words were directly contrasted, meaning processing of positive words and negative words are necessarily confounded with one another. Specifically, calculating the NAP effect for either positive or negative material would include the effects of inhibition for *both* positive and negative material. This prevents the priming effects of positive and negative words from being fully dissociated.

Modification to the NAP task have been made to address this concern by including neutral words. For example, one version [45] used neutral words as both potential targets and distractors to isolate the effects of ignoring either a positive or a negative word. However, trials where the prime-distractor was a neutral word were not analysed. This meant that control trials were still defined as trials where the probe-target valence is opposite to the prime-distractor valence. In other words, on a control trial, if the probe-target was negative, the prime-distractor was necessarily positive, whereas if the probe-target was positive the prime-distractor was inevitably negative. Under this design, it is unclear if the lack of a NAP effect can be attributed to inhibition of negative words. Instead, it may be driven by increased inhibition of positive words, with no change in inhibition of negative words.

Some tasks [48,51] have altered the control condition to use neutral material as the prime-distractor to address this limitation. One design [48] introduced neutral words as the prime and probe distractor for control trials ensuring the negative priming effects of positive and negative words were fully dissociated. However, participants in their experiment never responded to neutral words. Under this design, the contrast between positive and negative words is highlighted. The first effect is that the task is substantially easier, since there is very little opportunity for response ambiguity in the stimuli. More importantly, these dichotomous response alternatives might emphasize extreme emotions and cause more extreme evaluations (i.e. negative becomes more negative, positive becomes more positive). This subjective increase in the intensity of the response stimulus might artificially influence the amplitude of the NAP effect since emotional context has been shown to alter later, more evaluative processing of emotional material [34].

By using neutral words as both prime-distractors on control trials and as potential targets, attentional biases such as the negativity bias or positivity offset may be more effectively

examined. In such a design, the priming effects of positive and negative words are fully dissociated, which allows inhibitory processes for positive and negative material to be examined separately. Only one previous study [51] was found that incorporated both these features in a non-clinical sample. They reported an unexpected facilitation effect for *negative* material which contradicts previous work (e.g. [44,52]) that found a negative priming effect (i.e. slower responses on IR trials) for negative material in healthy controls. This facilitation effect is evidence of a negativity bias in healthy participants and may be attributable to a novel and more arousing study environment created by the fMRI scanner.

Using a similar design (called the modified NAP task here) in a less arousing study environment was expected to reveal a positivity offset. A study was designed wherein participants completed both the modified task as well as the original task [43]. This direct comparison is important to ensure that only the variations in the task accounts for any differences in the results. Congruent with previous results in healthy controls, negative priming was expected for both positive and negative words in the original NAP task; an effect that should be slightly larger for positive words than negative words. A different pattern of results was expected for the modified NAP task. Previously a categorization advantage has been reported for positive words [53]. In the modified NAP task, this was expected to take the form of a reduced NAP effect for positive words.

## Methods

Prior to data collection, the Simon Fraser University Department of Research Ethics approved this study (2013s0589). Participants provided written consent for data to be used.

**Participants.** Required sample size was calculated through an a-priori power analysis in G*Power 3.0.10. Expected effect sizes was estimated based on the nonsyphoric groups from three previous studies [43,45,52]. The average effect size for the difference between the NAP effect for positive and negative words was estimated to be a large effect ($dz$=.96). Importantly, the estimated effect size from [52] was substantially smaller ($dz$=.21). Thus, to account for potentially smaller effects within the modified NAP task, a conservative effects size equal to one third of the estimated average effect size ($dz$=.32) was used. This estimated the minimum sample size at 60 participants to detect a medium effect ($dz$=.32) with 80% power ($\alpha$=.05).

Sixty-two female undergraduate participants ($M_{Age}$=19.87 $SD$=1.97 $Min$=18 $Max$=27) with normal or corrected to normal vision received course credit via the online Research Participation System at Simon Fraser University. Since depressed symptoms modulate the NAP effect (see [52]), students reporting a history of affective disorder were excluded to avoid unnecessary biases in the results.

**Stimuli.** A pilot study was conducted to select the words for the present study. This was done to maximize the strength of the manipulation and reduce the influence of uncontrolled extraneous variables. A set of 80 positive (valence>6), 91 negative (valence<4), and 71 neutral (valence 4-6) words between 4 and 6 letters long were selected from the Affective Norms for English Words (ANEW: Bradley & Lang, 1999) database. In a categorization task, pilot study participants (N=9) indicated if each word was positive, neutral or negative. The 62 words within each category with the highest accuracy (>60%) and most consistent RTs were retained (Appendix A). In this final list, average word length did not differ between positive and negative words ($p$=.98), positive and neutral words ($p$=.50) or negative and neutral words ($p$=.51). Average arousal was similar for positive and negative words ($p$=.16). However, neutral words were significantly less arousing than both positive ($p$<.01) and negative words ($p$<.01). In terms of frequency ratings, the positive and neutral word lists did not differ significantly ($p$=.49) Frequency ratings for negative words was significantly lower than both positive

**Table 1. Linguistic information for the words used in the study.**

| Word Type | Valence | Arousal | Length | Frequency |
|---|---|---|---|---|
| Positive | 7.55 | 5.39 | 5.25 | 86.32 |
| Negative | 2.37 | 5.61 | 5.25 | 24.47 |
| Neutral | 5.28 | 3.91 | 5.16 | 102.51 |

(*p*<.01) and neutral words (*p*=.01; the implications of these significant differences are addressed in the discussion). Linguistic information on the words included in the study are provided in Table 1.

**Procedure and design.** After providing informed consent, participants were given some basic instructions about the experimental procedure and a medical and demographics questionnaire was administered. Additionally, two subscales (depression and positive emotions) from the NEO Personality Inventory – Revised (NEO-PI-R), were completed as part of a separate experiment. Finally, they completed two versions of the NAP task (standard and modified) in randomized order.

The NAP task was built and run in E-Prime 2.0.8 (Psychology Software Tools Inc.) on a Windows PC. Participants sat approximately 60 cm from a 19-inch LCD monitor (1024 x 786 resolution, 60 Hz refresh rate). During the task, coloured word pairs were presented for up to 2s, 1cm above and below a white fixation cross on a black background in size 18 Courier New font (letter size was .5cm). After a response was made or the time limit was reached, the central fixation cross was displayed on the screen alone for an interval of 500ms, preceding the next trial. The word colour (pink or yellow) indicated the target word, and the attended colour was counterbalanced between participants. On each trial, the position of the target (above or below the central fixation cross) was randomly assigned. Participants responded using the left arrow for a positive target, down arrow for a neutral target, and right arrow for a negative target. They were not informed of any differences between the prime and probe displays, and were simply instructed to respond to each trial display as quickly and accurately as possible.

To mimic the critical conditions of the original NAP task, six blocks were generated according to the following criteria: First, to control for any response repetition effects (see [54]) only IR and control trials were possible. Each of the four conditions (IR and control for each word valence) had to appear six times in each block. None of the words could be repeated within the same block, though the same words may appear in multiple blocks. And the order of the conditions was different in each block.

In the modified task, neutral words were included as both distractors and targets. To ensure that only IR and corresponding Control Trials were present in the experiment, no trials containing two words of the same valence were allowed. Thus, a positive and a negative word, a neutral and a positive word, or a negative and a neutral word could be presented together. Probe-target could not be preceded by a prime-target of the same valence (see Fig 1). Six blocks were generated according to the same criteria. However, the addition of neutral words meant there were six possible conditions (IR and Control for each word valence), and each could only appear five times per block.

For both tasks, blocks consisted of 30 trials, for a total of 360 experimental trials in each task. A 5s break between blocks and a 30s break between tasks was provided. During the shorter breaks, a visual reminder of the response options was provided, and more detailed instructions were given prior to each task. Any trials on which participants made an incorrect response, or with RT faster than 300ms were excluded from the analysis. No response was recorded after 2000ms.

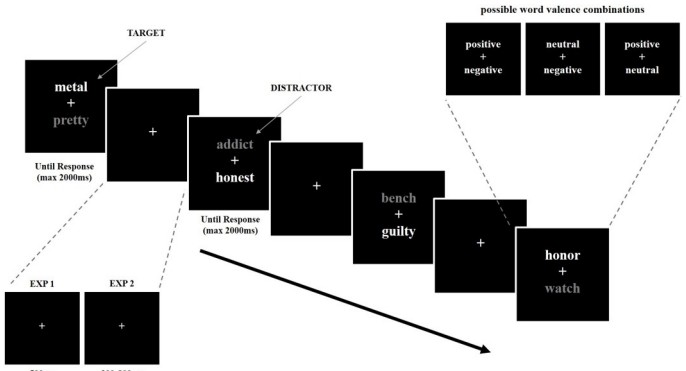

**Fig 1. Design of the modified negative affective priming task.**

**Statistical analysis.** One participant with extremely low accuracy (defined as below 33% for any word valence on the modified NAP task, or 50% for any word valence on the original NAP task) was excluded from the analysis.

Two (task: original, modified) by two (word valence: positive, negative) by two (condition: IR, control) repeated measures ANOVAs were conducted to compare accuracy and RT results. Within the RT analysis, the NAP effect was calculated as the difference between IR and control trials for each word valence to provide a measure of the cost associated with previously ignoring a word of the same valence as the current target. Follow-up t-tests were conducted to examine specific pair-wise comparisons. Family-wise error rates was controlled using a Bonferroni procedure.

Since neutral words in the modified task could not be included in these analyses, four t-tests were conducted to compare neutral words to positive and negative words in terms of accuracy and NAP results. Family-wise error rates was controlled using a Bonferroni procedure.

## Results

**Accuracy.** A significant main effect of task $F_{(1,60)}$=43.14, $\eta_p^2$=.42, $p$<.001, and trial type, $F_{(1,60)}$=9.47, $\eta_p^2$=.14, $p$=.003 emerged (Table 2). Accuracy was significantly higher in the original task than the modified task, $t_{(60)}$=6.56, $p$<.001, $d$=.76, suggesting that adding neutral words made the task more difficult. Accuracy was also higher on IR compared to control trials, $t_{(60)}$=3.08, $p$=.003, $d$=.16. A small, but significant two-way interaction between word valence and trial type $F_{(1,60)}$=4.08, $\eta_p^2$=.06, $p$=.05, revealed accuracy to negative targets were more slightly impacted by the prime-distractor. While accuracy for positive and negative targets was similar on IR trials, on control trials, accuracy to negative targets was slightly lower than positive targets.

In the modified task, accuracy for neutral words was significantly higher than both positive, and negative words, $t_{(60)}$=-2.36, $p$=.02, $d$=.28; $t_{(60)}$=-2.58, $p$=.01, $d$=.36. This suggests neutral words were easier to identify.

**Table 2. Average proportion of correct responses (standard deviation in parentheses).**

| Task | Positive Words | | Negative Words | | Neutral Words | |
|---|---|---|---|---|---|---|
| | **Ignored Repetition** | **Control** | **Ignored Repetition** | **Control** | **Ignored Repetition** | **Control** |
| **EXP. 1 OriginalNAP** | .95 (.09) | .94 (.08) | .94 (.08) | .93 (.08) | | |
| **EXP.1 ModifiedNAP** | .87 (.13) | .87 (.13) | .88 (.13) | .85 (.14) | .90 (.09) | .90 (.09) |
| **EXP. 2 ModifiedNAP** | .87 (.07) | .85 (.07) | .87 (.06) | .85 (.07) | .90 (.05) | .91 (.05) |

**Table 3. Average reaction time (standard deviation in parentheses), ms.**

| Task | Positive Words | | Negative Words | | Neutral Words | |
|------|----------------|--|----------------|--|---------------|--|
| | Ignored Repetition | Control | Ignored Repetition | Control | Ignored Repetition | Control |
| **EXP. 1 OriginalNAP** | 783.16 *(141.17)* | 762.96 *(124.53)* | 813.09 *(150.27)* | 803.08 *(143.03)* | | |
| **EXP.1 ModifiedNAP** | 869.89 *(130.28)* | 886.64 *(129.00)* | 912.85 *(141.42)* | 893.24 *(141.24)* | 918.64 *(161.35)* | 901.53 *(142.67)* |
| **EXP. 2 ModifiedNAP** | 950.02 *(116.04)* | 960.31 *(98.42)* | 981.71 *(116.73)* | 971.39 *(101.31)* | 966.10 *(115.03)* | 972.01 *(108.74)* |

**NAP RT.** The RT data (Table 3) revealed significant main effects of task, $F_{(1,60)}$=80.62, $\eta_p^2$=.573, $p$<.001, word valence, $F_{(1,60)}$=18.14, $\eta_p^2$=.232, $p$<.001 and trial type, $F_{(1,60)}$=5.66, $\eta_p^2$=.086, $p$=.021. Additionally, the three-way interaction was significant, $F_{(1,60)}$=9.32, $\eta_p^2$=.134, $p$=.003. These effects were driven by faster reaction times on IR trials for positive targets in the modified NAP task. Thus, when the NAP effect was calculated a significant *facilitation effect* for positive words in the modified NAP task emerged. The NAP effect to positive targets was significantly lower compared to and negative targets, in the modified task, $t_{(60)}$=-3.31, $p$=.002, $d$=.58, and also to positive targets in the original task, $t_{(60)}$=3.35, $p$<.001, $d$=.64.

Similarly, in the modified task, the NAP effect for neutral and negative words were comparable, $t_{(60)}$=.19, $p$=.85, $d$=.04. But a significant difference between the NAP effect for positive and neutral targets was observed, $t_{(60)}$=-2.86, $p$=.006, $d$=49.

## Discussion

Performance on the original NAP task in this study is congruent with that of the healthy controls in [43]. Thus, the differences in NAP scores observed in the modified NAP task cannot be attributed to unusual features of the present sample, or to other unforeseen methodological discrepancies.

Evidence for a positivity offset was observed in the modified NAP task. That is, a non-clinical sample in a neutral mood state showed a reduced NAP effect for positive words which took the form of a *facilitation* effect. This suggests that positive prime-distractors were not effectively ignored, which facilitated a subsequent response to a positive probe-target. This effect is congruent with other instances of the positivity offset. For example, a categorization advantage for positive words has been reported [53] and, a bias towards positive material has been described in normal (i.e. non-clinical) samples in the absence of a mood induction [55,56]. The present study expands this work by offering a possible mechanism underlying the positivity offset. Specifically, the facilitation effect suggests that inhibition for positive material is reduced in a neutral mood state, meaning task-irrelevant positive information is not effectively excluded from WM. But how this altered processing of the probe-target is still unclear. To better understand this, a second experiment using ERPs was conducted.

## Experiment 2

To the best of our knowledge, only two studies have used the NAP task in conjunction with ERPs. Unfortunately, neither provides information on the positivity offset due to designs optimized for comparing healthy controls and depressed patients and other methodological choices such as using faces for stimuli [44], or a design like the original NAP task [57]. However, since the latter study [57] used words as stimuli, the performance of their healthy control group might inform some expectations about the ERP modulations in the present study. They observed a larger amplitude P2 component to experimental trials for negative words over left and midline electrode sites, and a longer latency LPP over posterior parietal scalp. This offers two potential ERP components as indexes of attentional bias in the NAP task.

The P2 component described by [57] seems topographically and temporally similar to the EAP. Like the EPN, the EAP might reflect an early orienting bias to emotionally arousing stimuli [34]. The larger EAP to threat related words reported in anxious participants [24] may consequently be an artefact of the highly arousing experimental design. Thus, while a larger EAP may be observed to emotional words regardless of valence, no differences were expected between positive and negative words.

More crucially, the positivity offset is expected to manifest as a larger amplitude LPP to positive probe-targets. To maximize the positivity offset, a non-clinical sample in a neutral mood state responded to words as stimuli. Under these conditions, a larger amplitude LPP to positive words relative to negative words was observed in a word categorization task [16]. Consistently, we predicted that the amplitude of the LPP would be larger for positive words in the modified NAP task, accompanied by a pattern of RTs similar to experiment 1. Thus, a significant difference between the NAP effect for positive words and the NAP effect for negative words was anticipated.

## Methods

**Participants.**   Required sample size was calculated via an a-priori power analysis using G*Power 3.0.10. Previously, [26] reported a large effect size ($\eta_p^2$=.365) for the difference in LPP amplitude between neutral and emotional words, while [32] reported a substantially smaller effect ($\eta_p^2$=.15). To ensure the study was sufficiently powered, the latter effect size was used in the calculations ($\eta_p^2$=.15) which estimated the minimum sample size at 58 participants to detect the expected effect ($f$=.42) with 80% power ($\alpha$=.05).

Seventy-one female undergraduate students at Simon Fraser University ($M_{Age}$=18.75 $SD$=1.37 $Min$=17 $Max$=24) were recruited based on the same criteria as experiment 1. Participants with fewer than 30% of trials retained for any condition were removed from analysis. Thus, the final sample consisted of sixty individuals ($M_{Age}$=18.68 $SD$=1.36 $Min$=17 $Max$=24).

**Procedure and design.**   Participants completed the modified NAP task following the same procedures and design as in experiment 1 with three exceptions. First, to maximize the observed effect sizes, participants completed each of the six blocks twice. Second, to ensure that participants were indeed in a neutral mood state, they were asked to self-report their mood by answering the question "How do you feel?" on a 7-point Likert-type scale (anchors: 1-very sad, 4-neutral, 7-very happy) at the beginning and end of each set of six blocks. An average of these four ratings was taken to determine each participant's mood state throughout the experiment. On average mood ratings ($M$=3.97 $SD$=.44) did not differ significantly from a neutral rating of 4, $t_{(59)}$=-.51, $p$=.61. Finally, a jittered inter stimulus interval of 300-800ms was used to make the task suitable for ERP analysis.

**Electrophysiological recording.**   EEG activity was recorded using a sintered Ag/AgCl electrode cap with active electrodes at 64 standard Modified Combinatorial Nomenclature sites (Biosemi Active Two amplifier, Amsterdam). Additional electrodes were placed on the left and right mastoids, approximately 1cm lateral to the external canthi (measuring horizontal eye movements) and approximately 2 cm below each eye (measuring vertical eye movements and blinks). Voltage at each site was determined against a common mode sense (CMS) electrode and recorded at a sampling rate of 512Hz.

EEG for each participant was digitally filtered (0.01Hz high-pass, 30Hz low-pass, zero phase, 12dB/octave slope) and re-referenced to the average mastoid (BESA 5.3). Visual inspection and semiautomatic artefact rejection [58] were performed to eliminate trials containing blinks and eye movements or incorrect responses (Table 4). Distinct ERP averages were obtained for each of the 6 conditions (IR and Control for each word type) time-locked to word onset (200ms pre-stimulus baseline and 800ms post-stimulus).

**Table 4. Average number (standard deviation in parentheses) and proportion of trials retained in Experiment 2.**

| | Positive Words | | Negative Words | | Neutral Words | |
|---|---|---|---|---|---|---|
| | Ignored Repetition | Control | Ignored Repetition | Control | Ignored Repetition | Control |
| **Number of trials retained** | 42 (9.85) | 43 (8.21) | 44 (8.95) | 43 (8.21) | 45 (7.78) | 46 (8.79) |
| **Proportion** | .70 | .72 | .74 | .72 | .76 | .77 |

**Statistical analysis.** Three (word valence: positive, negative, neutral) by two (condition: IR, control) repeated measures ANOVAs were conducted to compare accuracy and RT results. Again, the NAP effect was calculated as the difference in RT between IR and control trials for each word valence. Follow-up t-tests were conducted to examine specific pair-wise comparisons. Family-wise error rates was controlled using a Bonferroni procedure.

ERP time windows were identified through visual inspection of the grand average waveforms and previously reported modulations by emotional words. Mean amplitudes were calculated between 190-260ms over a left anterior (electrodes F1, F3, F5) region of interest for the EAP and between 500-700ms post stimulus over a posterior (electrodes P3, P1, Pz, P2, P4) region of interest for the LPP.

The same three by two repeated-measures ANOVA was conducted for both the EAP and LPP. Family-wise error rates was controlled using a Bonferroni procedure.

## Results

**Behavioural.** In the accuracy data, significant main effects of word valence $F_{(2,118)}=13.12$, $\eta_p^2=.18$, $p<.001$, and trial type, $F_{(1,59)}=5.91$, $\eta_p^2=.09$, $p=.02$ emerged (Table 2). Accuracy was significantly higher to neutral than to positive, $t_{(59)}=-3.83$, $p<.001$, $d=.50$, or negative words, $t_{(59)}=-4.62$, $p<.001$, $d=.67$. Accuracy was also higher on IR compared to control trials, $t_{(59)}=2.43$, $p=.02$, $d=20$. A significant word valence by trial type interaction $F_{(2,118)}=7.99$, $\eta_p^2=.12$, $p=.001$, showed that accuracy to negative targets was slightly more impacted by the prime-distractor. While accuracy was similar on IR and control trials for both positive, $t_{(59)}=-4.62$, $p<.001$, $d=.15$, and neutral targets, $t_{(59)}=-4.62$, $p<.001$, $d=.18$, accuracy to negative targets was significantly lower on control trials compared to IR trials, $t_{(59)}=4.35$, $p<.001$, $d=.46$.

The RT data (Table 3) revealed a significant main effect of word valence, $F_{(2,118)}=6.20$, $\eta_p^2=.10$, $p=.003$, but no main effect of trial type, $F_{(1,59)}=.005$, $\eta_p^2<.001$, $p=.94$. When the NAP effect was calculated, the predicted facilitation effect was observed for positive words. Thus, although the two-way interaction was not significant, $F_{(1,59)}=2.67$, $\eta_p^2=.04$, $p=.07$, two t-tests comparing the NAP effect to positive words to that of negative and neutral words were conducted to test the specific a priori hypothesis. These showed that the NAP effect to positive targets was significantly lower compared to negative, $t_{(59)}=-2.11$, $p=.04$, $d=.34$, and neutral targets, $t_{(59)}=-1.98$, $p=.05$, $d=.35$.

**ERP.** Within the EAP time range (190-260ms) no significant effects were observed. Neither the main effect for word valence $F_{(2,118)}=.47$, $\eta_p^2<.01$, $p=.63$, nor the main effect for trial type, $F_{(1,59)}=1.29$, $\eta_p^2=.02$, $p=.26$, nor the 2-way interaction, $F_{(2,118)}=1.68$, $\eta_p^2=.03$, $p=.19$, were significant (see Fig 2).

Analysis of the LPP (500-700ms) showed significant main effects for word valence, $F_{(2,118)}=104.24$, $\eta_p^2=.64$, $p<.001$, and for trial type $F_{(1,59)}=19.24$, $\eta_p^2=.25$, $p<.001$. Relative to neutral targets ($M=1.74$, $SD=3.32$), both negative, ($M=5.79$, $SD=4.11$, $t_{59}=13.19$, $p<.001$, $d=1.08$), and positive targets ($M=5.42$, $SD=4.15$, $t_{59}=11.96$, $p<.001$, $d=.98$) showed a significantly larger LPP (Fig 3). Overall, IR ($M=4.83$, $SD=3.89$) compared to control ($M=3.81$, $SD=3.56$) trials, also elicited a larger LPP, $t_{59}=4.39$, $p<.001$, $d=.27$. The significant interaction,

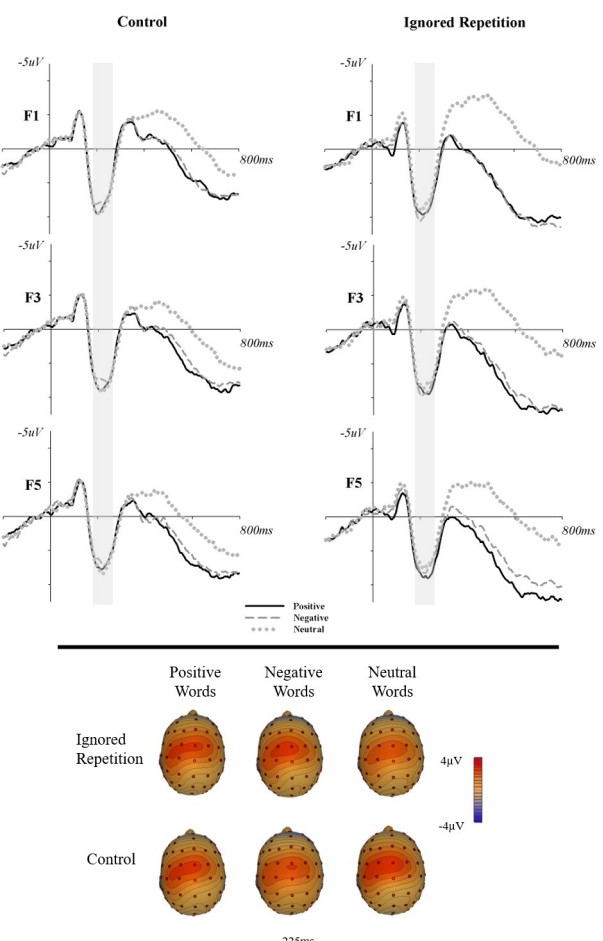

**Fig 2. Event-related potentials and topographic distribution of the early anterior positivity (EAP).**

$F_{(2,118)}$=5.03, $\eta_p^2$=.08, $p$=.008, was largely driven by positive words which showed a large and significant difference in mean amplitude between IR ($M$=6.38, $SD$=4.49) and control ($M$=4.47, $SD$=4.37) trials, $t_{59}$=4.71, $p$<.001, $d$=.43. A smaller difference between IR ($M$=6.20, $SD$=4.54) and control ($M$=5.38, $SD$=4.12) trials for negative words was also significant, $t_{59}$=2.31, $p$=.03, $d$=.19. Interestingly, the larger difference effect observed for positive words appears to be driven by a smaller LPP to positive, compared to negative targets on control trials, $t_{59}$=-2.27, $p$=.03, $d$=.21.

## Discussion

No differences were observed within the EAP time window which contradicts previous research reporting a larger amplitude EAP for threat-related targets [24]. Early emotion-related components such as the EAP and EPN may be sensitive to evolutionarily relevant stimuli [35]. Specifically, threat-related words may represent an especially arousing subcategory of negative words, and there may be an evolutionary advantage to responding to arousing negative stimuli quickly. In the present study, the average arousal rating between positive and negative words was controlled for and thus, no systematic effects between positive and negative words were observed.

The facilitation effect for positive words was accompanied by a large difference between the LPP amplitude on IR and control trials for positive words. Consistent with previous research

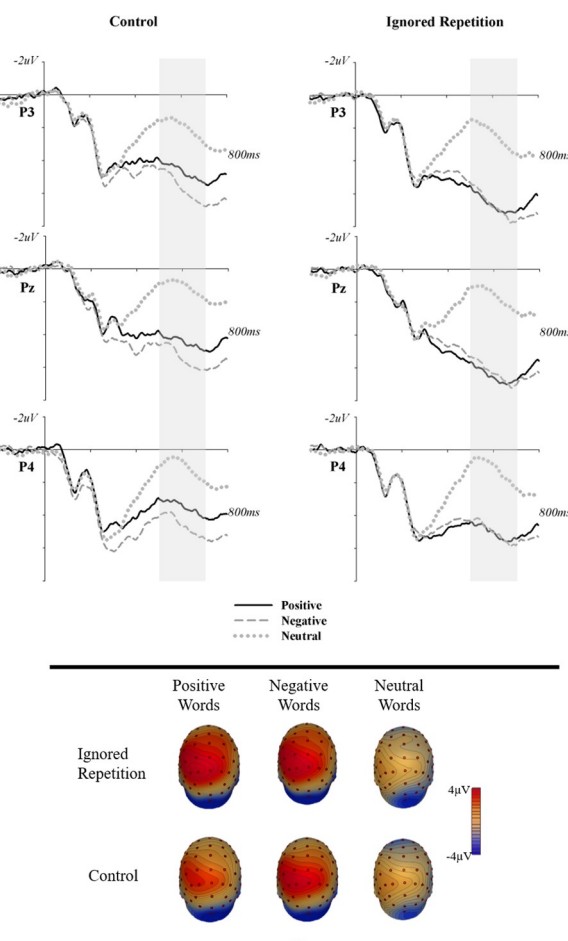

**Fig 3. Event-related potentials and topographic distribution of the late positive potential (LPP).**

[16,32,37], an LPP was observed to both positive and negative targets over centro-parietal scalp. And the amplitude of the LPP was larger on IR than Control trials. However, while a small but significant difference between IR and Control trials was observed for negative words, the difference was largest for positive words. Interestingly, positive words did not elicit a larger LPP than negative words on IR trials. Instead, LPP amplitude was significantly lower for positive words compared to negative words on Control Trials.

This suggests that, previously ignoring a word of the same valence as the current target (IR trials), may increase the motivational salience, and subsequent processing, of the target. The greater LPP amplitude for negative targets on Control Trials, suggests that negative material is generally more motivationally salient than positive targets. This is consistent with the negativity bias, and the threat-priority hypothesis [59] which suggests that there is an evolutionary bias developed out of a need to process potentially threatening stimuli quickly. The more interesting result is the large increase in LPP amplitude between IR and Control trials for positive words which indicates that previously ignoring a positive distractor increased the motivational salience of a subsequent positive target. This could explain the facilitation effect observed for positive targets since the increased processing would assist the entry and maintenance of positive material in WM, leading to faster responses on IR trials. These results are consistent with the broaden and build theory that suggests positive affect provides cues that the environment

is safe and facilitates engagement with the environment [13]. In this context, the positivity off-set represents an evolved mechanism to increase engagement with more elements in the environment. At low levels of arousal, the system increases positive evaluations of stimuli which increases the likelihood an individual may engage with that object. Results from the present study suggests that this may be accomplished by increasing the sensitivity of the selective attentional system to priming effects by positive material.

## General discussion

In the present study, a modified version of the NAP task was implemented to fully dissociate the negative priming effects of positive and negative words. The predicted difference between the NAP effect for positive and negative words was significant, but surprisingly took the form of a facilitation effect. Specifically, reaction times to positive targets was faster on IR than on Control trials. Traditional explanations of the NAP effect would suggest that participants failed to ignore the positive prime-distractors, which primed a subsequent response to a positive target. The larger LPP amplitude on IR trials compared to Control trials for positive words further suggest that previously ignoring a positive distractor enhanced the maintenance of a positive target in working memory. This is consistent with previous findings suggesting that the positivity offset is due to enhanced elaborative processing for positive words [16]. Thus, under conditions favouring the positivity offset (e.g. a non-clinical sample in a neutral mood state responding to low arousing stimuli), the processing of a positive target is facilitated by previous exposure to positive material. These results suggest that the positivity offset stems from reduced inhibitory processes at low levels of arousal.

Unexpectedly, LPP amplitude was greater for negative words than positive words on Control trials, while LPP amplitude was similar between positive and negative targets on IR trials. This suggests that negative targets are more motivationally salient in the absence of priming effects and may be better maintained in working memory. This could be explained by the threat-priority hypothesis [59]. One proposed mechanism for this effect is delayed disengagement which suggests that negative stimuli hold attention for longer [1] and delays other cognitive operations. As a result, RTs to negative material is generally slower than to positive or neutral material [19]. If this hypothesis is correct, then in the present study, attentional disengagement from the negative target was delayed which reduced the effect of previously experiencing a negative distractor compared to positive material.

It is worth noting that neutral words were significantly easier to identify correctly than negative and positive words in both experiments. This may have been due to a familiarity effect for neutral words. Frequency ratings were higher for neutral words than emotional words. Additionally, neutral words were used in control trials for both positive and negative words, meaning that they appeared more often as distractors than positive or negative words. While this may have affected the calculated NAP effects, it is not considered problematic for the comparison between positive and negative targets. Any familiarity effect of neutral prime-distractors would be constant for both positive and negative probe-targets and no differences in accuracy between positive and negative words were observed. Thus, none of the variance in RT on positive and negative words can be attributed to differences in difficulty.

Results from the modified NAP task used here and by [51] has important implications for the NAP task more generally. Research using the NAP paradigm has suggested that the negative ruminatory cycle in depression relies on insufficient inhibition of negative material [49]. Thus, the characteristic negative schemas [60] which causes the altered patterns in both memory and attention [61]) originates from an inability to effectively exclude and remove negative information from working memory [52,61]. However, because the priming effects of positive

and negative words are conflated in these previous studies, conclusions about altered processing of negative words only are not warranted. Instead, results from the modified NAP task used in this present study suggests that processing of positive words needs to be considered when interpreting NAP results from these studies.

## Limitations

A few limitations of the present study should be considered while interpreting these results. Firstly, the study was conducted on a female only, non-clinical sample. While this was a deliberate choice to reduce extraneous sources of variability, it limits the generalizability of the results. Future research should use the modified NAP task in clinical populations, to better understand the contribution of positive and negative material to depression separately.

Secondly, the finding that neutral words were easier to identify suggests that frequency needs to be more carefully controlled in future studies. In the present study, neutral words appeared more frequently as distractors, potentially making them more familiar to participants and thus, lowering the threshold for entering WM. Extending from this concern, linguistic frequency is a critical variable that impacts the processing of verbal stimuli at both behavioural and electrophysiological levels [62]. It is possible that words that appear more frequently in everyday use, have a similar advantage which may influence the allocation of selective attentional resources. In the current study, word frequency was found to vary significantly between negative words and both positive and neutral words. Unfortunately, during the design of the study it was determined that this limitation was unavoidable while accounting for other physical characteristics of word stimuli that might influence ERP effects (e.g., word length [62]), obtaining a reasonably even distribution of starting letters since (vowels and consonants have emotional connotations [63]), and avoiding words with multiple meaning. This challenge was further exacerbated by problem of how positive and negative words are used in everyday language. People tend to avoid negative emotions, and this is reflected in language use, thus, negative words are less frequent than positive and neutral words [64].

Our solution to the issue of word frequency in the current study was to perform the pilot study examining behavioural responses to the stimulus words. Results of this initial investigation ensured that there were no systematic accuracy or RT differences between participants' responses to positive, negative and neutral words, making us more confident that frequency and other extraneous variables were not exerting an undue influence on the results. However, [62] observed that frequency modulates ERP effects between 150-190ms and 320-360ms, and other studies such as [65] also show that word frequency affects earlier ERP components in the 150-200ms range. While these effects do not overlap with the LPP time-window in the present study (500-700ms post stimulus), some research shows that word frequency influences later ERP components as well. For example, word frequency modulates the amplitude of the N400 [66] and the interaction between word frequency and emotional valence modulated LPP amplitude [67]. Thus, it is possible that word frequency contributed to the LPP amplitude differences observed in the present study. This should be taken into account when considering these effects and results should be interpreted cautiously.

Thirdly, the words selected for this study belong to different grammatical classes which may have impacted the LPP results. Previous studies have shown that valence-driven modulations of later ERPs are most pronounced when the stimuli are adjectives [16,68]. This may be related to the more self-referential processing of adjectives since they are often used to describe a person's mood or personality [14]. In the present study, a combination of nouns and adjectives were used which may have attenuated the difference in LPP amplitude between positive and negative words. However, previous studies using both adjectives and nouns have observed

larger amplitudes in late ERP components to positive words [14,17] suggesting that the effects observed here are due to the emotional valence of the words and not their grammatical class.

Finally, the lack of EAP effects contradicts previous work that show the EAP is sensitive to threat-related words [24]. We suggest that this may be attributable to differences in the arousal qualities of the words used in the studies. Future research could use the modified NAP task to explore the influence of selective attentional mechanisms to anxiety in both clinical and non-clinical populations.

## Conclusion

Results from the present study extend previous research by offering a possible cognitive mechanism underlying the positivity offset. Specifically, positive material may benefit from reduced inhibition at low levels of arousal since irrelevant positive material may not be effectively prevented from entering working memory. This may increase elaborative processing of subsequent positive material.

## Supporting information

**S1 Fig. The complete lists of words used as stimuli in the study.**
(DOCX)

## Author Contributions

**Conceptualization:** Regard M. Booy.

**Data curation:** Regard M. Booy.

**Formal analysis:** Regard M. Booy.

**Supervision:** Patrick L. Carolan.

**Writing – original draft:** Regard M. Booy.

**Writing – review & editing:** Regard M. Booy.

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
