## [Decision Letter · Decision Letter 0]

13 Apr 2021

PONE-D-21-03837

The Role of Selective Attention in the Positivity Offset: Evidence from event related potentials.

PLOS ONE

Dear Dr. Booy,

Thank you for submitting your manuscript to PLOS ONE. I have received now a review from an expert and read the manuscript myself. While we both are generally positive about your manuscript, there are also a number of points that should be addressed in a revised version of the manuscript. Please, pay particular attention to the possible confounding effects of frequency (and you may like to take a view to a recent review about affective neurolinguistics in LC&N that discussed frequency effects on the processing of emotion words), which may require additional analyses (e.g., linear regression). 

We look forward to receiving your revised manuscript.

Kind regards,

José A Hinojosa, Ph.D.

Academic Editor

PLOS ONE

Journal Requirements:

Reviewers' comments:

Reviewer's Responses to Questions

**Comments to the Author**

1. Is the manuscript technically sound, and do the data support the conclusions?

Reviewer #1: Partly

2. Has the statistical analysis been performed appropriately and rigorously? 

Reviewer #1: Yes

3. Have the authors made all data underlying the findings in their manuscript fully available?

Reviewer #1: Yes

4. Is the manuscript presented in an intelligible fashion and written in standard English?

Reviewer #1: Yes

5. Review Comments to the Author

Reviewer #1: Review of the paper „The Role of Selective Attention in the Positivity Offset: Evidence from event related potentials”

The authors undertook the task of detailing the mechanisms involved in emotional stimuli encoding, which are related to attention orienting and selection depending on the level of arousal of differently valenced stimuli. These mechanisms are called negativity bias and positivity offset and are popular, yet difficult to grasp experimentally (in more specific terms) phenomena. Because of that the authors’ undertaking is a worthwhile one and moreover it is based on sound background. The paper showcases two experiments employing the NAP procedure with emotional words as stimuli (one behavioral, one psychophysiological) with well over a hundred participants, which is impressive in and of itself.

However, I have two major concerns and some minor ones that prevent me from fully recommending the paper for publication as of now.

Major concerns:

1. The stimuli were not matched on the word frequency value. The authors acknowledge this as a limitation of the study in the Discussion (in the context of higher accuracy rates of neutral words as compared to positive and negative ones). Frequency is one of the most important (if not THE most important) linguistic variables impacting and modulating the processing of verbal stimuli. Not matching it across subsets of the stimuli, not only limits the interpretation of the results obtained, but renders them uninterpretable (or makes completely different interpretation a distinct possibility). In other words we lose control over the independent variables. The differences in frequency value can explain not only the effect observed on the accuracy data as noted by the authors, but also the ERP data (the LPP result), since words of higher frequency were shown to elicit smaller amplitudes of the P3 family components (Hauk & Pulvermüller, 2004). The acknowledged difference in the frequency value between the neutral and emotional stimuli is a big enough problem, but there is also a noticeable difference between the positive and negative words themselves (Table 1, negative words being the least frequent)! Is that difference a significant one as well? I would like to see the statistical data on all three comparisons (neutral, negative, positive).

2. The low-arousing experimental conditions that the authors point to in the Introduction should be described there in a more clear manner and in more detail (they are stated clearly yet briefly in the Discussion though). Also, if possible, data on comparable studies employing a high-arousing conditions should be brought forward and briefly analysed (high arousing stimuli, mood induction studies) – this would serve as a better build-up for the hypotheses and help with the interpretation of the results. My main concern here though is the presupposed „neutral mood” of the healthy, non-clinical participants of the studies (including the present paper). This may not be so, since there are data that young, healthy participants are generally happy and usually in a moderately good mood (Diener & Diener, 1996; Herbert, Junghöfer, & Kissler, 2008). This could be evidenced in detail in a state of feeling good (hedonic tone), relaxed (tense arousal) and energetic (energy arousal, Matthews, Jones, & Chamberlain, 1990). So, the positivity bias results obtained in the study may be due to a kind of a positive mood congruence effect. The most important question is, did the authors measured the mood of the participants or gathered at least some feedback on their actual state? Were the subjects really in a neutral mood? As a possibility, the participants may have been in a good mood (hedonic tone) but in a low arousal state (very relaxed and rather tired), but only a good or proper, if you will, mood evaluation tool could tell us that!

Minor concerns:

1. The section of the Introduction related to the two ERP time windows of interest should be expanded and the possible processes these ERPs relate to better described. This concerns the EAP especially. More explanation and background is needed on the component, two of the three studies cited in this context in the Introduction do not mention the EAP at all. The authors should present more sources on the component (aside the paper by Taake et al., 2009). As far as the LPP is concerned, since the literature on it is abundant, the authors should present other interpretation of its role (alternative top-down theories). Also, the authors could add a paragraph on the selective attention itself and how it relates to the analysed ERPs and the NAP task.

2. The last sections of the Exp 2 Discussion merit further elaboration. The general discussion as a whole is lacking (e.g. should include more information on how to further apply the results of ERP studies and in broader terms, how they help us better understand selective attention in emotional word processing).

3. The words employed as stimuli belonged to different grammatical classes. I would like to see it mentioned and commented on in the general discussion.

4. I would like to see the time window analysed highlighted on both of the ERP figures.

Minuscule issues:

1. The citations in the text have some typos (missing colons, wrong year of the publication). The authors should glance through them all again.

2. I am not sure I understand the last sentence of the first paragraph on the page 12 (“They

responded to both the prime and probe displays and were unaware of any difference

between the two”).

3. Table 4 showcases the SD in parentheses, doesn’t it? Its caption should explain that.

References:

Diener, E., & Diener, C. (1996). Most people are happy. Psychological Science, 7, 181–185. https://doi.org/10.1111/j.1467-9280.1996.tb00354.x

Hauk, O., & Pulvermüller, F. (2004). Effects of word length and frequency on the human event-related potential. Clinical Neurophysiology, 115, 1090–1103. https://doi.org/10.1016/j.clinph.2003.12.020

Herbert, C., Junghöfer, M., & Kissler, J. (2008). Event related potentials to emotional adjectives during reading. Psychophysiology, 45, 487–498. https://doi.org/10.1111/j.1469-8986.2007.00638.x

Matthews, G., Jones, D. M., & Chamberlain, A. G. (1990). Refining the measurement of mood: The UWIST mood adjective checklist. British Journal of Psychology, 81, 17–42. https://doi.org/10.1111/j.2044-8295.1990.tb02343.x

6. PLOS authors have the option to publish the peer review history of their article (what does this mean?). If published, this will include your full peer review and any attached files.

Reviewer #1: No

---

## [Author Response · Author response to Decision Letter 0]

27 May 2021

Response to Reviewers

Dear Dr. Hinojosa,

We would like to thank you and the reviewer for your consideration of our paper. We have made changes to the manuscript in response to the concerns noted. These are described below, along with additional explanations addressing each of their concerns. 

We hope that these revisions, are well received. Changes in the revised manuscript are denoted in the file “SelectiveAttentionPositivityOffset_PLOS_Revised Manuscript with Track Changes” with blue and red font. Page numbers and paragraph information directing reviewers to the revisions noted below refer to the marked up manuscript. Thank you for your time and feedback. We look forward to your response.

Sincerely,

Regard Booy

Major concerns:

1. The first major concern the reviewer noted was that word frequency was not controlled for. We agree with the reviewer on the importance of word frequency, and to address this concern we have greatly expanded our discussion of this limitation within the manuscript (see pp. 28 – 29). Additionally, statistical tests of the differences in frequency between the three stimulus categories have been added on p. 13. The tests indicated that there was no significant difference between positive and neutral words (p=.49), but that negative words were significantly less frequent than both positive (p,.01) and neutral words (p=.01). Unfortunately, during the design of the study, it was determined that word frequency differences were unavoidable for practical reasons. When creating the list of words to be included in the study, it was not possible to control for all of the potentially important factors. We needed to take into account valence, arousal, word length, frequency and familiarity. We also needed to have a reasonably even distribution of starting letters since closed and open sounds have emotional connotations. And other semantic connotations further limited the available words. Obviously, words with multiple meanings had to be avoided, and words related to specific phobias could not be included, in order to avoid the possibility of strong personal associations of the words (see Herbert et. al. 2006). It was simply not possible to fully account for all of these variables and we had to prioritize controlling for valence, arousal and word length since these factors were anticipated to be the most likely to affect the ERP components of interest. 

There is a further problem of how positive and negative words are used in everyday language. People tend to avoid negative emotions, and this is reflected in language use. Thus, negative words are less frequent than positive and neutral words (Garcia, Garas & Schweitzer, 2012). This problem is exacerbated when trying to control for arousal, since high arousing negative words are especially infrequent. Thus, it proved impossible to build a list of words that controlled for valence, arousal and word-frequency. 

Our solution to maintain experimental control over the independent variable was to control for the behavioural effect of these extraneous variables by doing the pilot study. By ensuring that there were no systematic differences between participants’ responses to positive, negative and neutral words in terms of reaction time and accuracy, we could be more confident that extraneous variables like frequency were not exerting an undue influence on the results. 

Regarding the reviewer’s point that the frequency difference could explain the ERP results, we believe that this is unlikely. The word frequency effects noted by Hauk & Pulvermüller, (2004) occurred between 150-190ms and 320-360ms whereas the LPP time-window in the present study is between 500-700ms post stimulus. Other studies such as Strijkers, Costa & Thierry (2009) also show that word frequency affects earlier ERP components in the 150-200ms range. Thus, it is unlikely that differences in word frequency could account for the late component differences we observed. If word frequency did have an effect on later components, this would most likely have taken the form of a larger N400 to high frequency words (see Grainger, Lopez, Eddy, Dufau & Holcomb, 2012). In the present study, if the results were driven by word frequency, we would expect a more negative deflection to the higher frequency positive words, which is the opposite of what we observed. Thus, either the effects of valence is stronger than the effects of word frequency, or more likely, word frequency did not influence the amplitude of the LPP. This discussion has also been added to the limitations section (p. 29)

2. The reviewer’s second concern was regarding the neutral mood state of participants. In the present study, we are certain that participants were in a neutral mood. We did ask participants to rate their mood on a 7-point scale at four different times throughout the task. We then calculated an average mood for each participant. These ratings did not differ significantly from a neutral mood rating of 4. This should have been included in the original manuscript. We apologise for the oversight and thank the reviewer for catching it!

To address the reviewer’s concerns, the mood manipulation check data was added to the methods section of exp. 2 (see p. 20, “Procedure and Design). In short, there was no significant deviation from a baseline neutral mood in the study. We did not measure mood state in exp 1 since the primary purpose was to compare the different versions of the task. We also expanded the explanation of the low arousing conditions in the general introduction as the reviewer requested (p. 5).

Minor Concerns

1. As per the reviewer’s request we have expanded the description of the ERP components and the role of selective attention in the introduction (see p. 5 and 6). 

Regarding the reviewer’s comments about the EAP component, there is limited research on the component to date. Because it occurs in a similar time window as the EPN and is also elicited by emotional stimuli, it is thought to represent similar underlying cognitive effects. However, it is differentiated from the EPN by a different topography and polarity, but it is not yet clear if this is because these are merely opposite ends of a dipole, or if there are distinct neural generators for the two components. Further research is needed to differentiate these two components. We have included a brief discussion of this, and all of the research we are aware of on the EAP or similar components in the aforementioned revisions addressing this concern. 

2. We have elaborated on the potential implications of the LPP results in the discussion of experiment 2 (p. 25) and also expanded some relevant sections of the general discussion (p. 26, second paragraph) as requested by the reviewer. Additionally, see next.

3. The fact that words belong to different grammatical classes has been added as a limitation to the general discussion, and its possible impact on the LPP result is discussed (p. 30).

4. The time windows have been noted on the figures.

Miniscule Issues

1. We have double checked the in-text citations and made sure they are all correct. 

2. The last sentence of the first paragraph on the page 12 has been edited to be more explicit (now the last line of the first paragraph on p. 14 of the All Markup changes manuscript). 

3. The caption for table 4 has been updated as requested (p. 21). 

References

Garcia, D., Garas, A. & Schweitzer, F. (2012). Positive words carry less information than negative words. EPJ Data Science. 1(3). 

Grainger, J., Lopez, D., Eddy, M., Dufau, S., & Holcomb, P. J. (2012). How word frequency modulates masked repetition priming: An ERP investigation. Psychophysiology, 49(5), 604-616. https://doi.org/10.1111/j.1469-8986.2011.01337.x

Hauk, O., & Pulvermüller, F. (2004). Effects of word length and frequency on the human event-related potential. Clinical Neurophysiology, 115(5), 1090-1103. https://doi.org/10.1016/j.clinph.2003.12.020

Herbert, C., Kissler, J., Junghöfer, M., Peyk, P., & Rockstroh, B., (2006). Processing of emotional adjectives: Evidence from startle EMG and ERPs. Psychophysiology, 43, 197-206. doi: 10.1111/j.1469-8986.2006.00385.x.

Strijkers, K., Costa, A., & Thierry, G. (2010). Tracking lexical access in speech production: electrophysiological correlates of word frequency and cognate effects. Cerebral cortex, 20(4), 912-928. https://doi.org/10.1093/cercor/bhp153

---

## [Decision Letter · Decision Letter 1]

7 Jul 2021

PONE-D-21-03837R1

The Role of Selective Attention in the Positivity Offset: Evidence from event related potentials.

PLOS ONE

Dear Dr. Booy,

Thank you for submitting your revised manuscript to PLOS ONE. I have approached the original reviewer, who

felt that you have made a good job while addressing her/his previous concerns. However, there are still a few minor issues remaining (mainly related to word frequency effects), which you should further consider. Working myself in the field of affective language processing, I would like to recommend the authors to take into consideration recent reviews about the neural underpinnings of affective word processing, which have commented on word frequency effects (also in the LPP/LPC component). If the authors are willing to make these minor revisions, i am ready to make a final decision without sending the manuscript out for review.

We look forward to receiving your revised manuscript.

Kind regards,

José A Hinojosa, Ph.D.

Academic Editor

PLOS ONE

Journal Requirements:

Reviewers' comments:

Reviewer's Responses to Questions

**Comments to the Author**

1. If the authors have adequately addressed your comments raised in a previous round of review and you feel that this manuscript is now acceptable for publication, you may indicate that here to bypass the “Comments to the Author” section, enter your conflict of interest statement in the “Confidential to Editor” section, and submit your "Accept" recommendation.

Reviewer #1: (No Response)

2. Is the manuscript technically sound, and do the data support the conclusions?

Reviewer #1: Yes

3. Has the statistical analysis been performed appropriately and rigorously? 

Reviewer #1: Yes

4. Have the authors made all data underlying the findings in their manuscript fully available?

Reviewer #1: Yes

5. Is the manuscript presented in an intelligible fashion and written in standard English?

Reviewer #1: Yes

6. Review Comments to the Author

Reviewer #1: The authors have addressed most of my concerns and overall did a good job in revising the manuscript.

However, the unmatched frequency value between the negative and all the other word stimuli remains a major limitation of the study. The authors emphasize it in the revised version (in the discussion section), but I would encourage them to further elaborate on the issue and rewrite some of the paragraphs. Although the authors cite the studies that point to frequency having lesser impact on the later stages of processing, especially those marked by the LPP, it is still a distinct possibility, as was the case in the study by Palazova et al. (2011). And so I do not agree with the line of argumentation on pages 29-30 that states: “it is unlikely that differences in word frequency could account for the late component differences we observed. If word frequency did have an effect on later components, this would most likely have taken the form of a larger N400 to high frequency words (see Grainger, Eddy, Dufau & Holcomb, 2012). In the present study, if the results were driven by word frequency, we would

expect a more negative deflection to the higher frequency positive words, which is the

opposite of what we observed.”

The study by Grainger et al (2012) cited points to a complex pattern of N400 modulation based on repetition and frequency interaction for both high and low frequency words and the general frequency effect on the N400 (showcased in the study as well), to my knowledge, is exactly of the opposite characteristic, i.e. larger N400 to LOW frequency words (see also Dambacher et al., 2006; Halgren et al., 2002; Van Petten & Kutas, 1990). So there is a possibility that low-frequency value of the negative words as compared to the more frequent neutral and positive words had an impact on the results. The authors should consider such possibility in the discussion section, beside arguing strongly against it. Adding a phrase stating that because of the abovementioned issues, the results should be interpreted cautiously, would be welcome.

I understand that it was hard to prepare the stimuli and match them on all the important measures, especially when taking into account the emotional attributes. For future reference I recommend the authors use a huge normed database for verbal stimuli selection (the authors had used the ANEW dataset which consists of 1034 words), the largest being the extended ANEW (to almost 14000 words!), by Warriner et al. (2013).

I would also posit that frequency is far, far more important a factor to be controlled for than the distribution of starting letters which may evoke different connotations (we are not talking about bigram frequency here).

The authors have addressed all of my other points and I‘d like to express my gratitude and commend their efforts.

References

Dambacher, M., Kliegl, R., Hofmann, M., & Jacobs, A. M. (2006). Frequency and predictability effects on event-related potentials during reading. Brain Research, 1084(1), 89–103. https://doi.org/10.1016/j.brainres.2006.02.010

Halgren, E., Dhond, R. P., Christensen, N., Van Petten, C., Marinkovic, K., Lewine, J. D., & Dale, A. M. (2002). N400-like Magnetoencephalography Responses Modulated by Semantic Context, Word Frequency, and Lexical Class in Sentences. NeuroImage, 17(3), 1101–1116. https://doi.org/10.1006/nimg.2002.1268

Palazova, M., Mantwill, K., Sommer, W., & Schacht, A. (2011). Are effects of emotion in single words non-lexical? Evidence from event-related brain potentials. Neuropsychologia, 49(9), 2766–2775. https://doi.org/10.1016/j.neuropsychologia.2011.06.005

Van Petten, C., & Kutas, M. (1990). Interactions between sentence context and word frequencyinevent-related brainpotentials. Memory & Cognition, 18(4), 380–393. https://doi.org/10.3758/BF03197127

Warriner, A. B., Kuperman, V., & Brysbaert, M. (2013). Norms of valence, arousal, and dominance for 13,915 English lemmas. Behavior Research Methods, 45(4), 1191–1207. https://doi.org/10.3758/s13428-012-0314-x

7. PLOS authors have the option to publish the peer review history of their article (what does this mean?). If published, this will include your full peer review and any attached files.

Reviewer #1: No

---

## [Author Response · Author response to Decision Letter 1]

30 Sep 2021

Dear Dr. Hinojosa,

We would like to thank you and the reviewer for your consideration of our paper. We have made changes to the manuscript in response to the concerns noted. These are described below, along with additional explanations addressing each of their concerns. 

We hope that these revisions, are well received. Changes in the revised manuscript are denoted in the file “SelectiveAttentionPositivityOffset_PLOS_TrackChanges2” with red font. Page numbers and paragraph information directing reviewers to the revisions noted below refer to the marked up manuscript. Thank you for your time and feedback. We look forward to your response.

Sincerely,

Regard Booy

Major concerns:

1. The point of concern noted by the reviewer was the unmatched word frequency values between negative, neutral and positive words. Specifically, that frequency for negative words was significantly lower than both positive and neutral words. We would like to stress that we agree with the reviewer on the importance of word frequency, and we thank them for their careful consideration of this issue. 

The reviewer notes that we dismissed the possibility of word frequency effects on the later ERP components too readily and provides a number of studies showing the effect of low frequency on the N400, a component that temporally overlaps with the early part of our LPP window. It was not our intention to dismiss the importance of word frequency, but rather to emphasize the role emotional valence in the present study. We apologize that this was not evident and have revised the discussion of this issue accordingly. Specifically, as per the reviewer’s suggestion, in the discussion on p. 29 we note the possible influences of low frequency words on later ERP components and advise that the results be interpreted with caution.

---

## [Editor Report · Decision Letter 2]

4 Oct 2021

The Role of Selective Attention in the Positivity Offset: Evidence from event related potentials.

PONE-D-21-03837R2

Dear Dr. Booy,

We’re pleased to inform you that your manuscript has been judged scientifically suitable for publication and will be formally accepted for publication once it meets all outstanding technical requirements.

Kind regards,

José A Hinojosa, Ph.D.

Academic Editor

PLOS ONE
---

## [Editor Report · Acceptance letter]

25 Oct 2021

PONE-D-21-03837R2 

The role of selective attention in the positivity offset: Evidence from event related potentials. 

Dear Dr. Booy:

I'm pleased to inform you that your manuscript has been deemed suitable for publication in PLOS ONE. Congratulations! Your manuscript is now with our production department. 

Kind regards, 

on behalf of

Dr. José A Hinojosa 

Academic Editor

PLOS ONE